# Dietary magnesium intakes among women of reproductive age in Ghana—A comparison of two dietary analysis programs

**Helena J. Bentil** [1]*, **Seth Adu-Afarwuah**[2], **Joseph S. Rossi**[3], **Alison Tovar**[1], **Brietta M. Oaks**[1]

1 Department of Nutrition and Food Sciences, University of Rhode Island, Kingston, RI, United States of America, 2 Department of Nutrition and Food Science, University of Ghana, Legon, Accra, Ghana, 3 Department of Psychology, University of Rhode Island, Kingston, RI, United States of America

* helena_bentil@uri.edu

**Data Availability Statement:** All relevant data are within the paper and its Supporting Information files.

## Abstract

### Background

Despite the importance of magnesium to health and most importantly to women of reproductive age who are entering pregnancy, very few surveys have investigated the magnesium status of women of reproductive age, particularly in Africa. Additionally, the software and programs used to analyze dietary intake vary across countries in the region.

### Objective

To assess the dietary magnesium intake of women of reproductive age in Ghana and to compare the estimate of magnesium intake obtained from two commonly used dietary analysis programs.

### Methods

We collected magnesium intake from 63 Ghanaian women using a semiquantitative 150-item food frequency questionnaire. Dietary data was analyzed using two different dietary analysis programs, Nutrient Data Software for Research (NDSR) and the Elizabeth Stewart Hands and Associates (ESHA) Food Processor Nutrition Analysis software. We used the Wilcoxon signed rank test to compare the mean differences between the two dietary programs.

### Results

There were significant differences between the average dietary magnesium intake calculated by the two dietary programs, with ESHA estimating higher magnesium intake than NDSR (M±SE; ESHA: 200 ± 12 mg/day; NDSR: 168 ± 11 mg/day; $p<0.05$). The ESHA database included some ethnic foods and was flexible in terms of searching for food items which we found to be more accurate in assessing the magnesium intake of women in Ghana. Using the ESHA software, 84% of the study women had intake below the recommended dietary allowances (RDA) of 320mg/day.

**Funding:** Supported by startup funding awarded to Brietta Oaks from the University of Rhode Island, USA. The funders had no role in study design, data collection and analysis, decision to publish, or preparation of the manuscript.

**Competing interests:** The authors have declared that no competing interests exist.

## Conclusion

It is possible that the ESHA software provided an accurate estimate of magnesium in this population because it included specific ethnic foods. Concerted efforts such as magnesium supplementation and nutrition education should be considered to improve the magnesium intake of women of reproductive age in Ghana.

## Introduction

Magnesium serves as a cofactor in more than 300 enzymatic reactions including those responsible for regulating blood pressure, glycemic control and lipid peroxidation [1]. Several epidemiological studies and meta-analyses have reported inverse associations of dietary magnesium intake with the risk of cardiovascular disease, type 2 diabetes and hypertension [1–3]. In women of reproductive age, low magnesium intake during pregnancy is associated with a higher risk of restricted fetal growth, intrauterine growth restriction, gestational diabetes, preterm labor, and pre-eclampsia [4]. In addition, low magnesium intake during pregnancy has been identified as a main actor in the fetal programming of adult disease [4]. Specifically, intrauterine magnesium deficiency in the fetus could program insulin resistance after birth, inducing the increased risk of metabolic syndrome in adulthood [4].

Despite the importance of magnesium to health and most importantly to women of reproductive age who are entering pregnancy, very few surveys have investigated the magnesium status of women of reproductive age, particularly in Africa. Ghana is one of many countries in Africa experiencing the double burden of malnutrition which is defined as the presence of both under and overnutrition. Diabetes and cardiovascular disease are among the ten leading causes of death in Ghana in addition to neonatal disorders like prematurity [5] and of which inadequate magnesium intake has been linked to these health outcomes. According to data from 44 African countries to estimate per capita magnesium supply, the risk of dietary magnesium deficiency is low in most African countries, with the lowest risk of 0.5% in West Africa [6]. However, the analysis was based on per capita food supply which does not account for waste from cooking, spoilage, plate waste or household purchasing power and therefore, can overestimate national dietary consumption of magnesium. There is a need to assess the magnesium intake of Ghanaian women using a dietary assessment to determine the magnesium intake in this population.

Magnesium status is usually determined through assessments of dietary magnesium intake. Serum magnesium is a poor indicator of total body magnesium stores, accounting for only 0.3% of total body magnesium [7]. In addition, serum magnesium is maintained at a relatively constant level even during periods of low intake of magnesium with intracellular magnesium in the bone, and muscles serving as a reservoir to stabilize serum concentration when intake is low [7, 8]. Although dietary assessment is the most reliable way to determine magnesium status, there is variation in the software and programs used to analyze dietary intake across countries. For example, dietary studies in West African countries, often use the ESHA (Elizabeth Stewart Hands and Associates) Food Processor [9] while studies in other countries, including the U.S., typically use the NDSR (Nutrient Data Software for Research) because of its associated food composition database (Nutrition Coordinating Center (NCC) Food and Nutrient Database) [10]. Currently, there are no guidelines or consensus on which software should be used and it is also unknown whether they provide differing estimates of magnesium intake. Given that magnesium status is best assessed via diet and there is a variation in the software

used to analyze dietary intake, there is a need to compare estimates of magnesium calculated using different dietary software.

Therefore, the goal of this study was to investigate the magnesium status in women of reproductive age in Ghana and to compare estimates of magnesium calculated using ESHA and NDSR.

## Materials and methods

### Study design, setting and participants

Between July and August 2019, we conducted a pilot cross-sectional study among 63 women living in Odumase Krobo, a peri-urban area and the district capital of Lower Manya Krobo District in the Eastern Region of Ghana, located about 70 km north of the capital city, Accra. The primary purpose of the study was to examine the association between magnesium intake and biomarkers of diabetes risk. Main results and a detailed description of the sampling process has been published elsewhere [11]. Women were recruited into the study if they were between the ages of 18–49 years old, a resident of Odumase Krobo, able to speak either Krobo, Twi, or English and non-pregnant according to self-report. Eligible women were visited at their homes. During the home visits, women were consented and completed an interviewer-administered survey to collect information on socio-demographic characteristics, a food frequency questionnaire (FFQ) to collect information on dietary magnesium intake, and height and weight measurements. The study was approved by the Ghana Health Service Ethical Committee (REF# GHS-ERC016/10/18) and the University of Rhode Island Institutional Review Board (REF# BI1819-005). Written informed consent was obtained from all study participants before participation. Participants agreed to participate by either signing or thumbprinting the informed consent form.

### Socio-demographic variables and anthropometry

We used an interviewer-administered questionnaire programmed using the REDCap data collection tool to collect data on socio-demographic characteristics including age, ethnicity, marital status (single, married or separated/widowed/divorced), number of successful pregnancies, completed educational level (none, primary education or less, high school, or tertiary level), employment status (employed or not employed), sources of drinking water and type of sanitation facility. We classified sources of drinking water and type of sanitation facility as improved or unimproved based on WHO's classification to determine the proportion of women who had access to improved vs. unimproved water and sanitation [12]. Households that use piped water connected in dwelling, plot or yard; public standpipes; boreholes; protected dug wells; protected springs and rainwater collection were classified as having an improved source of water. If a household sanitation facility included either a flush or pour-flush connected to a public sewer or septic system, sewer system, septic tank pit latrines, ventilated-improved pit latrines, or pit latrines with slab or composting toilets, it was considered an improved source. Unimproved sanitation included shared or public-use sanitation facilities and pit latrines without slabs or open pits, bucket/hanging latrines, or open defecation.

We measured height and weight using standard procedures [13] with a height board (UNICEF S0114540) and digital weight scale (SECA 874) respectively. Both measurements were taken with the women barefoot and wearing light clothes. Height was recorded to the nearest 0.1 cm and weight was recorded to the nearest 0.1kg. Body mass index (BMI) was calculated as weight (kg) / (height (m))$^2$ and classified according to the World Health Organization criteria [12]: underweight (BMI $<$ 18.5 kg/m$^2$), normal weight (BMI 18.5–24.9 kg/m$^2$), overweight (BMI 25.0–29.9 kg/m$^2$) or obese (BMI $\geq$ 30.0 kg/m$^2$).

### Assessment of dietary magnesium intake

Dietary magnesium intake were assessed using a semiquantitative 150-item food frequency questionnaire (FFQ) which was interviewer administered and has been used in larger studies and national surveys in Ghana [13–15]. For each food item, we assigned a portion size using standard household measures such as cups, tablespoons, teaspoons, glasses as well as using photographs from a Food Amounts Booklet [16]. Study participants were asked how often on average they consumed that amount in the past week. The frequency of consumption of specified portion size was asked in six categories: never, once per week, 2 times per week, 3–4 times per week, 5–6 times per week, once per day, and 2 or more times per day. We estimated daily magnesium intake by multiplying the frequency of consumption of each food item by the magnesium content of the specified portion size.

The Nutrient Data Software for Research 2018 (NDSR; University of Minnesota, Minneapolis, MN, USA) and the ESHA Food Processor Nutrition Analysis software, version 10.8 (ESHA Research Inc, 2010, Salem, Oregon) were used to estimate dietary magnesium intake. For composite and mixed dishes, recipes were added to both the ESHA Food Processor and NDSR Nutrition Analysis software. In both programs, the United States Department of Agriculture (USDA) dietary database was used as the primary reference because it is comprehensive and regularly updated. The magnesium content of specific Ghanaian foods not available in either the ESHA Food Processor or NDSR were obtained from the Food and Agricultural Organization West African Food Composition table.

We also collected information on dietary supplement intake. However, none of the reported dietary supplements contained magnesium and were therefore not included in the analysis.

### Statistical analyses

Categorical variables are expressed as frequencies and percentages and as mean ± standard deviation for continuous variables. Comparisons between ESHA and NDSR dietary magnesium estimates were performed using a Wilcoxon signed rank test, the non-parametric version of a paired samples t-test. All analyses were performed in the SAS Studio 5.2 (SAS Institute, Cary, NC, USA). Statistical analysis was two-tailed and a p-value < 0.05 was considered statistically significant. Assuming a modest effect size of one-third of a standard deviation difference between the means and a modest correlation between the ESHR and NDSR measures of 0.60, the expected statistical power of the Wilcoxon test is 0.811.

## Results

### Socio-demographic and anthropometry characteristics

Socio-demographic characteristics and anthropometry of the study participants are presented in Table 1. The mean (SD) age of our study participants was 29.5 (8.5) years. Of the respondents, 56% were single, separated or widowed. Approximately 98% were of the Ga-Dangme ethnic group; 22% had completed senior high school and above; 62% were employed; 78% had been pregnant (successful) at least once; 62% had improved source of drinking water and 86% had unimproved sanitation facilities. More than half of the study participants (44%) were overweight/obese.

### Average dietary magnesium intake

Fig 1 presents the comparisons between the two dietary analysis programs. There was a significant difference between the estimated magnesium intake between the two programs, with a

**Table 1. Socio-demographics and anthropometry status of study participants.**

| Variables | Frequency (%) or Mean ± SD |
|---|---|
| Age, years | 29.5 ± 8.5 |
| Marital Status | |
| Married | 28 (44.4%) |
| Single/Separated/Widow | 35 (55.6%) |
| Ethnicity | |
| Ga-Dangme | 62 (98.4%) |
| Others | 1 (1.6%) |
| Educational Background | |
| None | 7 (11.1%) |
| KG/Primary/JHS (Low) | 42 (66.7%) |
| SHS/tertiary (High) | 14 (22.2%) |
| Employment status | |
| Not employed | 24 (38.1%) |
| Employed | 39 (61.9%) |
| Number of pregnancies | |
| None | 14 (22.2%) |
| ≥1 | 49 (77.8%) |
| Source of drinking water | |
| Improved | 39 (61.9%) |
| Unimproved | 24 (38.1%) |
| Sanitation facility | |
| Improved | 9 (14.3%) |
| Unimproved | 54 (85.7%) |
| Mean BMI, kg/m$^2$ | 25.2 ± 5.1 |
| BMI Category | |
| Underweight | 1 (1.6%) |
| Normal | 34 (54.0%) |
| Overweight/ Obese | 28 (44.4%) |

KG = Kindergarten, JHS = Junior High School, SHS = Senior High School, BMI = Body Mass Index

mean (SE) and median magnesium intake of 200 (12) mg/day and 185 mg/day for ESHA and 168 (11) mg/day and 145 mg/day for NDSR [mean difference (MD) = 32.23; 95% CI = 16.80, 47.65; P <0.0001]; and a Spearman correlation of 0.67. Both programs showed that a majority of the participants did not meet their recommended daily allowance for magnesium (ESHA: 84.1%; NDSR: 96.8%). Fig 2 presents the major contributors to magnesium intake. These were *banku* made from fermented corn, *fufu* and *kokonte* both made from cassava, *koose* made from cowpeas, *hausa koko* made from millet, light soup made from eggplant, smoked tuna, plantain, yam, and orange.

## Discussion

We aimed to assess the dietary magnesium intake of Ghanaian women of reproductive age and also to compare the estimates of magnesium calculated using ESHA and NDSR dietary analysis programs. The average dietary intake of magnesium was below recommended dietary allowances with the majority of the women not meet the RDA of 320 mg/day. This is concerning for women because a low magnesium status at the start of pregnancy increases the risk of developing pre-eclampsia, gestational diabetes, preterm labor, restricted fetal growth, and

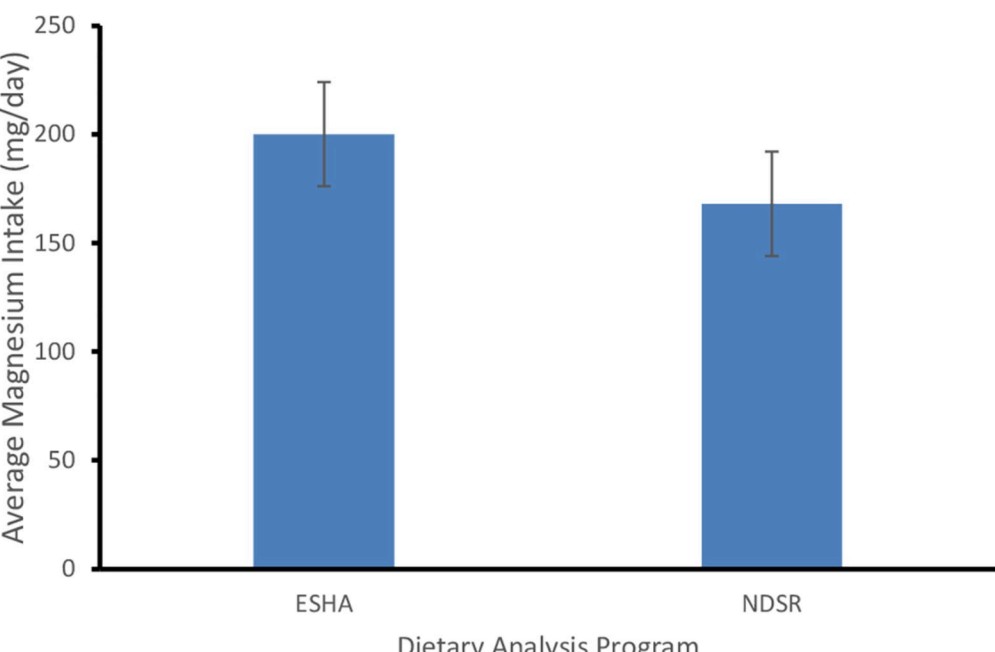

**Fig 1. Average magnesium intake by ESHA and NDSR dietary programs.**

intrauterine growth restriction [4]. We also observed significant differences between the dietary magnesium intake assessed by the dietary analysis programs.

Low magnesium intakes have been reported in similar populations in Ghana and other countries. Frimpong, et al. [17] assessed the dietary intakes of bank employees in Ghana using

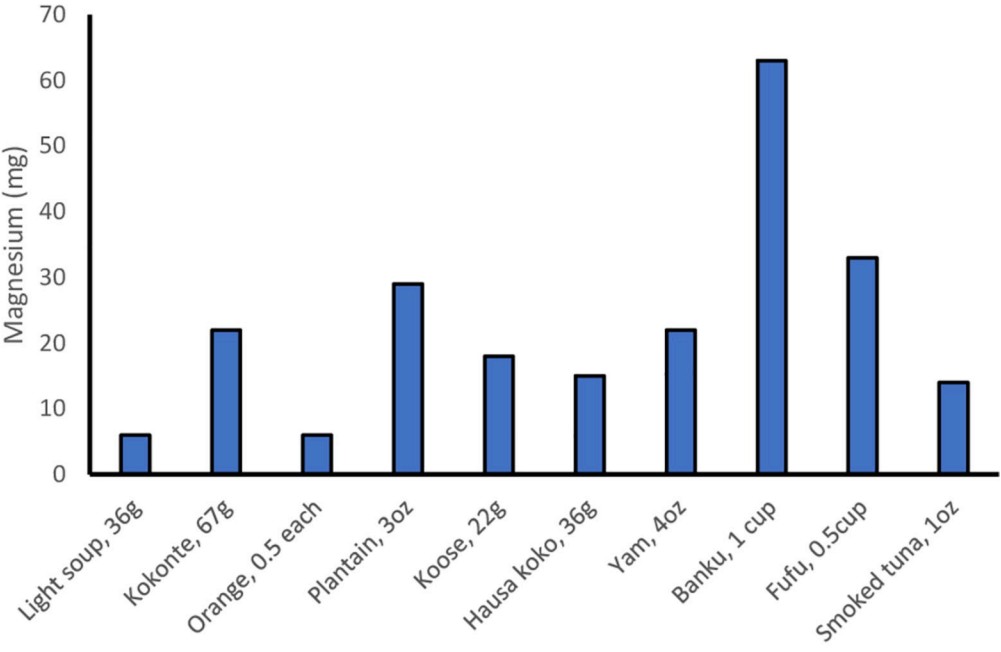

**Fig 2. Ten food sources of magnesium consumed by the study participants.**

ESHA and reported a mean dietary magnesium intake of 88 mg/day for women, which was lower than our results. The difference could be attributed to differences in the dietary patterns of the urban population in Frimpong's study versus the peri-urban dietary patterns of our study participants, as well as the employment status of the participants, which in Frimpong's study included bank employees, whereas our study participants, while mostly employed, were mostly involved in unskilled or skilled labor such as hairdresser and trader. Joy et al. [6], on the other hand, found in their study that countries in West Africa had a mean regional supply of magnesium of 1,019 mg per capita per day based on their assessment of the country's specific food balance sheets. In their study, they found that the grain, sorghum, which is an excellent source of magnesium, contributed greatly (559 mg per 100 g) to the region's magnesium supply. However, sorghum was not one of the reported grains consumed by our study participants which could most likely account for the large differences. The magnesium food sources in our study ranged from animal to plant sources, but the majority came from grains like corn and millet, as well as starchy roots and tubers like cassava and yam. However, none of these sources provided 559 mg per 100g. For example, 100 g (half a US cup) of *banku* made from corn contains 33 mg of magnesium.

The suboptimal magnesium intakes observed in our study and other populations could be due in part to the low consumption of whole grains, legumes, fruits and vegetables that is seen in Western diets. Consistent with our finding, in the United States, for example, the reported average magnesium intake of women is 228mg/day [18]. Ford et al. [19] found similar intakes in three ethnic groups in the United States: 256 mg/day among Caucasian women, 202 mg/day among African American women, and 242 mg/day among Mexican American women. The Shanghai Women's Health Study in China reported an average daily intake of 267 mg [19]. In the German population, women's magnesium intake was found to be 200 mg respectively [20]. Dietary patterns in Ghana like in other low-and middle-income countries have gradually been westernized due to urbanization and economic growth [21]. The western diet is often low in magnesium due to the refining and processing of foods [1]. Refined and processed foods contain almost no magnesium [1]. For example, wheat refined to flour, rice to polished rice, and corn to starch deplete magnesium by 82%, 83%, and 97%, respectively [8]. Magnesium is mainly found in unprocessed foods like green vegetables, whole grains, legumes and nuts [1].

Both the ESHA Food Processor and NDSR programs have been used to estimate nutrient intakes in several populations [17, 22–27]. We selected ESHA Food Processor as the principal software in our study since it is regularly used in West Africa to assess magnesium intake [9], but we also wanted to compare the results to other systems, such as NDSR, which is widely used in the United States and there is currently no formal guidance as to which should be used in Ghana [10]. The ESHA food processor has been used in several studies especially in West Africa because of the inclusion of certain ethnic foods like fufu in the database not found in the NDSR database. This may have accounted for the differences in the estimates of magnesium calculated using the programs. The search of food and beverage items in the ESHA food processor is easy and one may not necessarily have to be specific in searching for food and beverage items in the database. However, searching for food and beverage items in NDSR is a bit more complicated because one must search by food groups. When food groups differ between countries, it becomes more complicated. For example, in West Africa, tomatoes are classified as vegetables, but in the United States, they are classified as fruits. In addition, while potatoes are classified as a vegetable in the United States, they are not in West Africa. Moreover, the cost of the programs varies. The cost of the ESHA food processor software is less expensive and the subscription can be canceled at any time. For NDSR, an initial software license has to be procured making it more expensive. This makes the ESHA food processor more accessible in terms of cost making it affordable for many researchers in Africa. Both programs have the

feature to add recipes for composite dishes and to add a new ingredient not found in the database. This feature makes it possible to add new ingredients from other sources like the West African Food Composition database which we used in the present study. The West African food composition table came out in 2012 and was developed by the Food and Agriculture Organization of the United Nations (FAO) in collaboration with the West African Health Organization (WAHO) of the Economic Community of West African States (ECOWAS), Bioversity International, and the International Network of Food Data Systems (INFOODs) [28]. It is available at no cost and includes a composition of 472 foods from the Western Africa region [28].

Of note, food composition databases in Africa either do not exist or are not kept up to date. Of the 54 countries in Africa, only 22 have any kind of food composition databases with 9 countries having a national comprehensive food composition database [29]. A systematic review of how dietary data is analyzed in Africa showed that several countries in Africa either lack a country-specific food composition database or do not have an updated one which often results in them using food composition tables available in the region or the global database [9]. Examples of cited databases the systematic review found included the USDA Nutrient Database for Standard Reference and food composition tables developed by Food and Agriculture Organization (FAO), such as the West African Food Composition Table [9]. Ghana has no national comprehensive food composition table but has two publications of nutrient composition of some commonly consumed foods [29]. These publications were published in the years 1977 and 1983 and have since not been updated. Similarly, publications available in Togo (1957), Senegal (1961), Cameroon (1966) and Mali (1998) have not been updated since the years they were first published [29]. Nigeria is the only West African country that has a national food composition database which was published in 2019 [29]. The Nigeria Food Database 2019 is an open-access digital platform that is available online. While this is exciting to see for Nigeria, we do need to see it for all West African countries. Although regional and global food composition databases are useful resources to provide estimates of foods and beverages in the absence of a country-specific food composition database, it is imperative that countries have their own up-to-date food composition databases as some nutrients in foods are dependent on soil and feed which varies by region and country.

Our study had several limitations. First, we recognize that the food items, their quantity and frequency of consumption reported by participants may have been influenced by recall bias which is an inherent problem of using the FFQ dietary assessment method. This could have led to underreporting and overreporting of food intakes. Second, because Ghana has no comprehensive national food composition database the USDA food composition database and FAO West African Composition table was used which could be below or above the reported consumed magnesium intake. Third, we had a small sample size and our target population was recruited from a particular region in Ghana which limits generalizability. However, our study has several strengths to note. This is one of the first studies focusing specifically on magnesium intakes of women of reproductive age in Africa. In addition, we estimated magnesium intakes using two dietary analysis programs. Lastly, while we acknowledge the limitations of the FFQ method, we note that our FFQ was comprehensive in capturing magnesium intake because it included both food and dietary supplement sources of magnesium.

## Conclusion

This study offers valuable information on the magnesium intake of women of reproductive age in Africa which indicates that the majority are not meeting the RDA for magnesium. This calls for concerted efforts to improve the magnesium intake of women of reproductive age as

this would lay the foundation for their productivity as well as for the health of future generations. Although the in our analyses we found that the ESHA software provided an accurate estimate of magnesium in this population because of its inclusion of specific ethnic foods, we would like to emphasize the importance of African countries having their own up-to-date food composition databases.

## Supporting information

**S1 File. Inclusivity in global research.**
(PDF)

**S1 Dataset.**
(XLS)

## Acknowledgments

We thank Mr. Isaac Baah Sackitey for his assistance with data collection. We thank all the study participants.

## Author Contributions

**Conceptualization:** Helena J. Bentil, Joseph S. Rossi, Alison Tovar, Brietta M. Oaks.

**Data curation:** Helena J. Bentil.

**Formal analysis:** Helena J. Bentil.

**Funding acquisition:** Brietta M. Oaks.

**Investigation:** Helena J. Bentil.

**Methodology:** Helena J. Bentil, Seth Adu-Afarwuah, Joseph S. Rossi, Alison Tovar, Brietta M. Oaks.

**Project administration:** Helena J. Bentil, Brietta M. Oaks.

**Resources:** Brietta M. Oaks.

**Software:** Brietta M. Oaks.

**Supervision:** Joseph S. Rossi, Alison Tovar, Brietta M. Oaks.

**Writing – original draft:** Helena J. Bentil.

**Writing – review & editing:** Seth Adu-Afarwuah, Joseph S. Rossi, Alison Tovar, Brietta M. Oaks.

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
