## [Decision Letter · Decision Letter 0]

26 Sep 2022

PONE-D-22-23183Dietary magnesium intake among women of reproductive age in Ghana - A comparison of two dietary analysis programsPLOS ONE

Dear Dr. Bentil,

Thank you for submitting your manuscript to PLOS ONE. After careful consideration, we feel that it has merit but does not fully meet PLOS ONE’s publication criteria as it currently stands. Therefore, we invite you to submit a revised version of the manuscript that addresses the points raised during the review process.

We look forward to receiving your revised manuscript.

Kind regards,

Linglin Xie

Academic Editor

PLOS ONE

Journal Requirements:

"Supported by startup funding awarded to Brietta Oaks from the University of Rhode Island, USA"

5. Please amend either the title on the online submission form (via Edit Submission) or the title in the manuscript so that they are identical.

6. Please ensure that you refer to Figure 2 in your text as, if accepted, production will need this reference to link the reader to the figure.

Reviewers' comments:

Reviewer's Responses to Questions

**Comments to the Author**

1. Is the manuscript technically sound, and do the data support the conclusions?

Reviewer #1: Partly

2. Has the statistical analysis been performed appropriately and rigorously? 

Reviewer #1: I Don't Know

3. Have the authors made all data underlying the findings in their manuscript fully available?

Reviewer #1: No

4. Is the manuscript presented in an intelligible fashion and written in standard English?

Reviewer #1: Yes

5. Review Comments to the Author

Reviewer #1: Reviewer’s results for the article of PONE-D-22-23183

Major comments

This is the article studying daily magnesium intake using different two methods of FFQ. The authors showed that one of the methods seems better to that of another. In ither words, ESHA seems superior to NDSR because ESHA involves West African Food database for magnesium. However, my concern is that in the methods part, the authors state that the magnesium content of specific Ghanaian foods not available in either ESHA or NDSR were obtained from the Food and Agriculture Organization West African Food Composition table (lines 146 – 8). From this description, magnesium content could be obtained both to ESHA and NDSR. If so, if the responder responded food A which is not involved both, the authors could obtain magnesium content to both and there might not be different between ESHA and NDSR.

The other small concerns are as the follows:

1. the readers could not understand which foods are related with magnesium intake. In the other words, there are no descriptions which food are more frequently taken to explain to magnesium intake are more. To explain this, the authors must show the relationship of most of daily magnesium intakes as nutrient with food which had taken most.

2. No information of frequencies of foods intakes to explain magnesium intake.

3. The authors showed that daily magnesium intake using ESHA and NDSR were significantly different, but as far as the figure 2 and standard deviation (SD) of both described in the article in lines 179 and 180, the difference between the two is not seen significant different because of large SDs of them.

4. No information of pregnancy outcomes with magnesium deficiency is described because the outcome of magnesium deficiency must be critical issue in this article. For resolve this, the authors must show the relationship between magnesium deficiency and pregnancy adverse outcomes using the objective outcome parameters.

The authors must describe the matters to the abovementioned issues. Then, this article might be accepted after the further reviewing.

Minor comments

1. The tables to show the relationships between daily intake of foods and magnesium to explain which foods are most to contribute to magnesium deficiencies.

6. PLOS authors have the option to publish the peer review history of their article (what does this mean?). If published, this will include your full peer review and any attached files.

Reviewer #1: No

---

## [Author Response · Author response to Decision Letter 0]

6 Mar 2023

Response to editor’s comments

1. Thank you for stating the following financial disclosure:

[Supported by startup funding awarded to Brietta Oaks from the University of Rhode Island, USA].

2. Your ethics statement should only appear in the Methods section of your manuscript. If your ethics statement is written in any section besides the Methods, please move it to the Methods section and delete it from any other section.

This has been deleted.

This has been deleted.

---

## [Decision Letter · Decision Letter 1]

5 Apr 2023

Dietary magnesium intakes among women of reproductive age in Ghana - A comparison of two dietary analysis programs

PONE-D-22-23183R1

Dear Dr. Bentil,

We’re pleased to inform you that your manuscript has been judged scientifically suitable for publication and will be formally accepted for publication once it meets all outstanding technical requirements.

Kind regards,

Linglin Xie

Academic Editor

PLOS ONE

Additional Editor Comments (optional):

Reviewers' comments:

Reviewer's Responses to Questions

**Comments to the Author**

1. If the authors have adequately addressed your comments raised in a previous round of review and you feel that this manuscript is now acceptable for publication, you may indicate that here to bypass the “Comments to the Author” section, enter your conflict of interest statement in the “Confidential to Editor” section, and submit your "Accept" recommendation.

Reviewer #1: All comments have been addressed

2. Is the manuscript technically sound, and do the data support the conclusions?

Reviewer #1: Yes

3. Has the statistical analysis been performed appropriately and rigorously? 

Reviewer #1: Yes

4. Have the authors made all data underlying the findings in their manuscript fully available?

Reviewer #1: Yes

5. Is the manuscript presented in an intelligible fashion and written in standard English?

Reviewer #1: (No Response)

6. Review Comments to the Author

Reviewer #1: Major comments

This is the article to show an accuracy of novel developed software (ESHA) to analyze daily intake of magnesium using gold standard of NDSR. The authors showed its usefulness because of including ethnic foods in ESHA. However, ESHA could not utilized for the other ethnicities and it must be limitation of ESHA. However, this article could show the availability of novel developed software for analysis of ethnic foods in the world.

As the result, this could be accepted for readers in worldwide.

7. PLOS authors have the option to publish the peer review history of their article (what does this mean?). If published, this will include your full peer review and any attached files.

Reviewer #1: No

---

## [Editor Report · Acceptance letter]

24 Apr 2023

PONE-D-22-23183R1 

Dietary magnesium intakes among women of reproductive age in Ghana - A comparison of two dietary analysis programs 

Dear Dr. Bentil:

I'm pleased to inform you that your manuscript has been deemed suitable for publication in PLOS ONE. Congratulations! Your manuscript is now with our production department. 

Kind regards, 

on behalf of

Dr. Linglin Xie 

Academic Editor

PLOS ONE